# The Impact of Training on the Loss of Cardiorespiratory Fitness in Aging Masters Endurance Athletes

**DOI:** 10.3390/ijerph191711050

**Published:** 2022-09-03

**Authors:** Johannes Burtscher, Barbara Strasser, Martin Burtscher, Gregoire P. Millet

**Affiliations:** 1Department of Biomedical Sciences, University of Lausanne, CH-1015 Lausanne, Switzerland; 2Institute of Sport Sciences, University of Lausanne, CH-1015 Lausanne, Switzerland; 3Medical Faculty, Sigmund Freud Private University, A-1020 Vienna, Austria; 4Department of Sport Science, University of Innsbruck, A-6020 Innsbruck, Austria

**Keywords:** aerobic exercise capacity, exercise training, training cessation, training (re)uptake, aging

## Abstract

Elite masters endurance athletes are considered models of optimal healthy aging due to the maintenance of high cardiorespiratory fitness (CRF) until old age. Whereas a drop in VO_2_max in masters athletes has been broadly investigated, the modifying impact of training still remains a matter of debate. Longitudinal observations in masters endurance athletes demonstrated VO_2_max declines between −5% and −46% per decade that were closely related to changes in training volume. Here, using regression analyses, we show that 54% and 39% of the variance in observed VO_2_max decline in male and female athletes, respectively is explained by changes in training volume. An almost linear VO_2_max decrease was observed in studies on young and older athletes, as well as non-athletes, starting a few days after training cessation, with a decline of as much as −20% after 12 weeks. Besides a decline in stroke volume and cardiac output, training cessation was accompanied by considerable reductions in citrate synthase and succinate dehydrogenase activity (reduction in mitochondrial content and oxidative capacity). This reduction could largely be rescued within similar time periods of training (re)uptake. It is evident that training reduction or cessation leads to a considerably accelerated VO_2_max drop, as compared to the gradual aging-related VO_2_max decline, which can rapidly nullify many of the benefits of preceding long-term training efforts.

## 1. Introduction

Highly functional older individuals in general [1] and elite masters endurance athletes in particular are considered models of optimal healthy aging [2], which is characterized by the maintenance of a high level of cardiorespiratory fitness (CRF) in older age. CRF can be assessed by individual maximal aerobic power (VO_2_max) [3,4]. Ample evidence attests a close association between CRF and endurance exercise performance with longevity [5,6,7,8]. Conversely, a decline in CRF is related to reduced endurance performance and an elevated risk of morbidity and mortality [9,10]. A landmark study by Saltin and colleagues [11] convincingly demonstrated a rapid loss of CRF when healthy and physically active individuals became physically inactive (i.e., 20 days of bed rest). A relatively fast recovery and an increase in VO_2_max above baseline was observed during a subsequent 8-week training period [11]. Accordingly, high CRF, even if acquired and maintained through regular exercise training for many years (i.e., a typical condition for masters athletes), rapidly drops when training is ceased, e.g., due to injury or illness, jeopardizing the hard-earned exercise benefits [12]. Whereas the aging-related decline in VO_2_max has been comprehensively evaluated in general and endurance-trained populations [6,13,14], the importance of maintaining appropriate training loads and, in particular, the consequences of short- and long-term training reduction/cessation and those of training (re)uptake in masters endurance athletes have been much less investigated. However, regular and appropriate training stimuli are of utmost importance to improve or maintain muscular performance and CRF [11,12]. Thus, here, we aim to evaluate and summarize longitudinal observations on VO_2_max changes related to training variations of masters endurance athletes in the long and short term.

## 2. Materials and Methods

The present narrative review is reported following the IMRAD (introduction, methods, results, discussion) format [15]. A comprehensive search and screening strategy was used to identify relevant literature. PubMed and Web of Science were searched using combinations of terms covering the topics of training-related VO_2_max changes in masters endurance (primarily runners) athletes: (masters OR master OR endurance athletes OR elite runners) AND (VO_2_max OR aerobic power OR aerobic capacity) AND (humans) AND (aging OR older) AND (training). Studies that assessed VO_2_max changes longitudinally and reported associated changes in at least one parameter of training load were considered eligible for inclusion. We focused on studies that followed participants over several time points and/or evaluated participants of different age groups.

In addition, we performed a literature search on studies reporting effects of short-term (up to 12 weeks) training cessation or training (re)uptake (beginning with training or training reuptake after training cessation) on VO_2_max and associated physiological changes. Because such data on masters athletes are scarce, a representative range of studies including young and older healthy subjects (initial VO_2_max > 25 mL/min/kg) was considered, with a focus on studies that followed participants over several time points.

Data are presented descriptively. Multiple linear regression (stepwise variable selection) was applied to evaluate the predictive importance of independent variables with respect to VO_2_max changes. Training volume, age, sex, the observation period (in years) and baseline VO_2_max (at the beginning of the longitudinal observation) were considered as independent variables.

## 3. Results

Results of six long-term studies that satisfied our inclusion criteria on masters athletes reporting longitudinal changes in VO_2_max with aging and related changes of training characteristics are shown in Table 1.

The number of the predominantly male participants in the various age groups ranged between 6 and 34. The age of participants after follow-up varied between 46.5 and 82.8 years, and the observation periods ranged between 6 and 22 years (Table 1). The VO_2_max (mL/min/kg) decline per decade ranged from −5% to −46% per decade and was closely related to the changes in training volume (running, km/week) (Table 1, Figure 1).

Studies reporting training intensities (running, min/km) indicate that training volume reductions parallel intensity reductions, except one of the female groups of a study by Eskurza et al., who increased training volume but reduced training intensity during follow-up [16]. Male masters who maintained near-normal training volume (not more than 10% reduction or maintenance of high training volume) showed a VO_2_max decline of between −5% and −6.5% per decade [17,18,19,20,21].

**Table 1 ijerph-19-11050-t001:** Longitudinal changes in VO_2_max with aging and related changes of training characteristics in masters endurance athletes. Positive values of the training intensity measure (min/km) indicate a decrease in intensity.

Reference	N m (Males) f (Females)	Observation Period (Years)	Age (Years) Post	VO_2_max (mL/min/kg) Pre vs. Post (% Change) Change Per Decade	Training V: Volume, km/Week I: Intensity, min/km Pre vs. Post (% Change)
Eskurza et al., 2002 [16]	6 f	6	61.0	45.2 ± 2.1 vs. 42.1 ± 2.1 (−7%) −12%	V: 38.0 vs. 45.8 (+20) I: 5.3 vs. 5.6 (+5%)
Eskuzra et al., 2002 [16]	10 f	6	56.0	50.0 ± 2.2 vs. 43.8 ± 2.2 (−12%) −20%	V: 62.0 vs. 42.0 (−32%) I: 5.3 vs. 5.6 (+5%)
Hawkins et al., 2001 [21]	31 m	9	53.5	58.7 ± 1.7 vs. 50.4 ± 1.5 (−14%) −16%	V: 61.8 vs. 43.6 (−29%)
Hawkins et al., 2001 [21]	34 m	8	62.2	53.4 ± 1.4 vs. 46.2 ± 1.4 (−13%) −17%	V: 56.2 vs. 43.3 (−23%)
Hawkins et al., 2001 [21]	13 m	9	71.1	46.2 ± 2.5 vs. 36.4 ± 2.6 (−21%) −23%	V: 43.8 vs. 37.5 (−14%)
Hawkins et al., 2001 [21]	8 m	7	82.8	41.5 ± 3.1 vs. 28.4 ± 2.7 (−32%) −46%	V: 49.4 vs. 26.7 (−46%)
Hawkins et al., 2001 [21]	24 f	8	51.2	48.7 ± 1.6 vs. 45.2 ± 1.2 (−7%) −9%	V: 55.1 vs. 37.7 (−22%)
Hawkins et al., 2001 [21]	16 f	8	58.3	46.7 ± 1.3 vs. 40.8 ± 1.8 (−13%) −16%	V: 39.4 vs. 31.8 (−19%)
Hawkins et al., 2001 [21]	9 f	8	73.2	39.4 ± 1.6 vs. 31.8 ± 2.8 (−19%) −24%	V: 43.6 vs. 26.6 (−39%)
Pollock et al. 1997 [19]	9 m	9.2	60.4 ± 8.5	55.4 ± 8.7 vs. 52.1 ± 6.8 (−6%) −6.5%	V: 61 vs. 55 (−10%) I: 4.9 vs. 5.2 (+6%)
Pollock et al. 1997 [19]	9m	10	70.4 ± 8.5	52.1 ± 6.8 vs. 43.2 ± 6.3 (−17%) −17%	V: 55 vs. 35 (−36%) I: 5.2 vs. 5.9 (+14%)
Pollock et al. 1997 [19]	10 m	10	59.5 ± 10.3	54.2 ± 7.7 vs. 50.0 ± 6.9 (−8%) −8%	V: 49 vs. 38 (−12%) I: 4.9 vs. 5.4 (+10%)
Pollock et al. 1997 [19]	10 m	10	69.8 ± 10.2	50.0 ± 6.9 vs. 40.8 ± 9.5 (−18%) −18%	V: 38 vs. 27 (−29%) I: 5.4 vs. 6.3 (+17%)
Katzel et al. 2001 [20]	7 m	8.7	70	51.3 ± 2.4 vs. 48.6 ± 1.8 (−5%) −6%	highly trained (no essential change)
Katzel et al. 2001 [20]	21 m	8.7	71	49.8 ± 1.1 vs. 38.2 ± 0.9 (−23%) −26%	moderately trained (volume and intensity reduction)
Katzel et al. 2001 [20]	12 m	8.7	74	49.4 ± 2.2 vs. 33.8 ± 1.8 (−32%) −36%	not trained (rather sedentary)
Rogers et al. 1990 [18]	15 m	8	62	54.0 ± 1.7 vs. 51.8 ± 1.8 (−4%) −5%	highly trained
Trappe et al. 1996 [17]	10 m	22	46.5	68.8 vs. 59.2 (−14%) −6%	highly trained
Trappe et al. 1996 [17]	18 m	22	46.5	64.1 vs. 48.9 (−24%) −11%	moderately trained
Trappe et al. 1996 [17]	15 m	22	46.5	70.7 vs. 46.7 (−34%) −15%	not trained

Those who reduced training volume by between 11% and 20% or were moderately trained showed a VO_2_max decline of between −8% and −26% per decade [17,18,19,20,21], and those with training volume reductions of more than 20% or who became almost sedentary had a VO_2_max decline of between −15% and −46% per decade [17,18,19,20,21]. Eskurza et al. and Hawkins et al. reported data on female athletes, also indicating a training-dependent loss of VO_2_max [16,21]. Regression analysis including male athletes [17,18,19,20,21] revealed a close association between reported VO_2_max reductions and related changes in training volumes with aging. Fifty-four percent of the variance in the observed VO_2_max decline was explained by training-volume changes (Figure 1), and this percentage increased to 70% when the age of the athletes was considered. No other variables improved the explanation of the VO_2_max decline. Within the groups of females [16,21], 39% of the variance in VO_2_max change was explained by changes in the training volume (Figure 1).

**Figure 1 ijerph-19-11050-f001:**
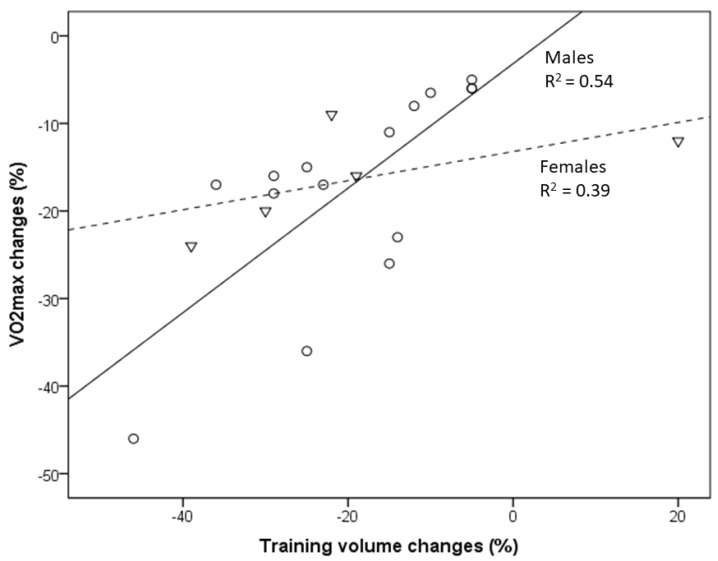
Relationship between VO_2_max decline and the reduction in training volume with aging of masters athletes (from data presented in Table 1) [16,17,18,19,20,21]. Circles indicate males, and triangles indicate females. Changes in the training volume explain 54% and 39% of the variance in VO_2_max changes in male and female masters athletes, respectively.

Few data on training cessation or training (re)uptake responses (physiological characteristics) in older masters are available. Thus, results of studies on young and older endurance athletes and healthy individuals reporting short-term changes in VO_2_max and related physiological changes due to training cessation or training (re)uptake were considered for this analysis (Table 2).

Despite some variations, an almost linear decrease in VO_2_max up to about −20% from a few days to 12 weeks of training cessation was demonstrated [22,23,24,25,28,29,30,32,34,35,36,38] (Table 2 and Figure 2). Based on the few studies on females [24,25], no important sex difference could be derived. However, when training cessation is associated with bed rest, a marked decline in the VO_2_max (e.g., −16.5%) may occur after only 3 days, which seems to be more pronounced in well-trained compared to rather sedentary individuals [37]. With training (re)uptake, VO_2_max again increases considerably in healthy young and older individuals, but this increase was larger during a 12-week training period in older male subjects [31] and one female masters athlete [12] (29.3% and 31.2%) when compared to young individuals (15.4%) (Table 2 and Figure 2). The observed decrease in VO_2_max was primarily associated with a reduction in cardiac output (decrease in stroke volume with even slight increases in maximal heart rate), especially shortly after training cessation, but a diminished arteriovenous oxygen difference became increasingly important with increasing duration of training cessation (Table 2 and Figure 2). In addition, training cessation was accompanied by relatively large and increasing reductions in citrate synthase and succinate dehydrogenase activity (up to about −40%) in 12 weeks. Citrate synthase is a commonly used marker for mitochondrial content and succinate dehydrogenase activity, providing information about the capacity of mitochondrial complex II, a protein complex contributing to oxidative phosphorylation, as well as ATP production and cellular energy supply. A reduction in these two markers indicates a reduced cellular aerobic capacity based on reduced mitochondrial content and/or efficiency. With training (re)uptake in older individuals, continuously increasing cardiac output with training duration contributed more importantly to the VO_2_max improvement than the arteriovenous–oxygen differences (Table 2). This is in contrast to young individuals, who showed a larger improvement of the arteriovenous–oxygen difference, especially during the first weeks of training (Table 2).

## 4. Discussion

The presented findings from longitudinal studies highlight the importance of continuing training activities with respect to VO_2_max changes of masters athletes in the long and short term. Based on studies including young and older athletes, as well as non-athletes, a considerable percentage of age-related VO_2_max decline (per decade) seems to occur rapidly (e.g., within 12 weeks) after training cessation. However, this effect can be largely restored by appropriate training (re)uptake. Changes in cardiac output (more importantly) and arteriovenous oxygen difference accompanied by reduced levels and activity of mitochondrial enzymes (e.g., citrate synthase and succinate dehydrogenase) are associated with training cessation or training (re)uptake interventions.

### 4.1. VO_2_max Decline in the Long Term: Effects of Training in Masters Athletes

Masters endurance athletes represent an excellent model of healthy aging that is relevant for the evaluation of consequences of changing in training habit on VO_2_max. Although VO_2_max seems to inexorably decrease with aging, even in life-long runners, the potential modulatory capacity of short- and long-term training variations remains a matter of debate. In healthy sedentary adults of both sexes, VO_2_max declines by about 10% per decade after the age of 25–30 years and slightly more at older ages (e.g., older than 60–70 years). Overall, this rate seems not to differ much in endurance athletes [6]. Longitudinal data from a healthy population indicate an exacerbated VO_2_max reduction with aging; whereas the reduction was about 8% in the fourth decade (over 10 years), it amounted 23% in the seventh decade and was more pronounced in males than females [13]. Whereas higher physical activity levels were associated with higher VO_2_max values across all ages and in both sexes, physical activity (quartiles) did not change the slope of the VO_2_max drop [13]. However, this may not hold true for (at least male) masters endurance athletes, who are able to maintain high training volumes (and likely intensity) until old age (Table 1, Figure 1). The presented data derived from longitudinal studies are in line with cross-sectional studies [39]. Cross-sectional data from 203 men (male endurance athletes and untrained subjects, 20–90 years) indicate that the weekly training volume constitutes a significant positive predictor of age-related changes in aerobic capacity [40]. Other cross-sectional data of masters athletes (20 power and 19 endurance athletes, 37–90 years) recently reported a peak aerobic and anaerobic power decline of around 7–14% per decade, with no difference between athletic disciplines or sexes [41]. Because exercise tests were stopped when heart rates exceeded the age-predicted maximal heart rate by 10 bpm and due to the lack of information on training characteristics, assessments of potential training effects on the observed VO_2_max decline are not possible [41]. Similarly, another cross-sectional study suggests that active older men (n = 146) lose maximal aerobic power at a rate of about 12%, and active older women (n = 82) lose maximal aerobic power at a rate of about 8% per decade between the ages of 40 and 80 years [42]. However, the VO_2_max decline becomes considerably steeper in masters athletes older than 80 years old, even in the absence of major illnesses or orthopedic issues [43]. Again, aging and training effects cannot be readily disentangled in that study due to the large variations regarding years of training (15.2 ± 9.7) and weekly training milage (33.5 ±19.3) [42].

In a comparison of American road running events, it was recently shown that the sex gap in performance decreases with increased age [44]. In the present study, the association between VO_2_max decline and changes in training volume (and intensity) is less pronounced in female compared to male endurance athletes [16,21]. This may partly be related to the lack of studies evaluating female athletes but also to the outlier of one female group studied by Eskurza et al. [16]. Nevertheless, a somewhat lower trainability of aerobic capacity in women may contribute to those differences [45].

Longitudinal changes in VO_2_max in cross-country skiers (age 58.7 ± 2.3 years) who continuously competed and reported unchanged training patterns amounted to only −4.1 ± 3.7 % per decade [46], which fits well with longitudinal data from male masters athletes (runners) presented in the present study, with a decrease in VO_2_max of between −5% and −6.5% per decade shown in those who continued to engage in regular vigorous endurance exercise, training and competitions [17,18,19,20]. The exaggerated VO_2_max decline in masters athletes who reduced training volumes (Figure 1) can be readily explained from both training scientific and practical perspectives [47,48,49]. Training characteristics (e.g., training volume, intensity and frequency) are undisputedly important modifiers of VO_2_max in both young and older athletes [50]. Reductions in volume and/or intensity of exercise training and the resulting decline in VO_2_max and performance in aging endurance athletes may be caused by various factors, such as decreasing motivation to train and/or compete, musculoskeletal injuries, pain and/or aging-related diseases [2,6,51,52,53], sometimes even associated with excessive training and competition activities [54]. Consequently, the aging-related CRF decrease, especially in male masters athletes, will remain small in those who are able to avoid factors provoking considerable training reductions. In summary, male masters athletes who can maintain their training volume (and intensity) can minimize their VO_2_max decline to about −5% to −6.5% per decade [17,18,19,20], but those values increase to up to −26% in those who reduce their training to a moderate level and can even be as high as −46% per decade in those who become sedentary [17,18,19,20,21].

### 4.2. Is There a Potential Role for Body Composition Explaining the VO_2_max Decline in Masters Athletes?

The effects of regular exercise are closely related to body composition and are therefore difficult to disentangle. Aging-related training reductions or cessation are often associated with changes in body composition. Longitudinal and cross-sectional data [55] indicate a significant contribution of lean body mass (LBM) alterations to VO_2_max changes, an effect that seems to be more pronounced in males than in females. Accordingly, a recent study demonstrated that lifelong aerobic exercise attenuated the age-related loss of quadricep muscle volume by about 50% in men but not in women, whereas adipose tissue infiltration into muscle was attenuated by about 50% in both sexes [56]. Higher training intensity throughout life provided increased protection against adipose tissue infiltration into muscle. The aerobic qualities of skeletal muscle among these lifelong exercisers likely contributed to the strong muscle mass-to-VO_2_max relationship. In line with these finding, no association between appendicular muscle mass and age was found in highly active older adults of both sexes [57]. The significant difference in muscle mass between exercising and sedentary individuals clearly illustrates that levels of physical activity play a considerable role in the phenotype of aging muscle. A recent study provided novel data on body composition in masters track and field athletes [58]. The authors concluded that loss of skeletal muscle mass and changes in bioimpedance phase angle (a raw parameter of cellular function and an indicator of tissue hydration and nutritional status [59]) are important contributors to the age-related reduction in anaerobic power, even in athletes who maintain high levels of physical activity in old age, indicating a deterioration of muscle quality in old age. However, it is also possible that a favorable body composition and nutritional conditions are prerequisites to participate in masters sport. Therefore, future research employing longitudinal study designs is needed, with the aim of determining which of these measures alone or in combination optimally predict individuals’ fitness and general health markers.

### 4.3. VO_2_max Decline in the Short Term: Effects of Training Cessation and Training (Re)Uptake

The VO_2_max decline in endurance-trained individuals usually does not occur gradually over decades but rather consists of a “slow component” related to aging per se and a “rapid component” due to training reduction/cessation. As demonstrated in the presented results, an almost linear VO_2_max decrease or increase was observed during a 12-week period of training cessation or training (re)uptake, respectively, in healthy young and older individuals [12,24,28,31]. Compared to the −5% to −6.5% VO_2_max decline within a decade in male masters athletes [17,18,19,20], VO_2_max steeply decreased as a result of training cessation, even in young endurance-trained (the majority of participants included in this study were male) athletes by −7% after only 12 days and dropped further to −18% after 12 weeks of training cessation [22]. Similar observations have been published for young competitive female runners, for whom 5 days of training cessation did not yield significant changes, whereas 10 days resulted in a 7.6% drop, which was linked to reduced running performance after 15 days [24].

However, VO_2_max rapidly improves after a few weeks of training (re)uptake after cessation or when training starts from a sedentary status. VO_2_max improvements occurred more rapidly during the first 3 weeks (12% vs. 8.4%) in young compared to older healthy men, but after 12 weeks, the increases were even higher in the older compared to younger men (15.4% vs. 29.3%) [31], at least in the selected studies considered here (Figure 2). A 31% VO_2_max increase was reported in a female masters athlete after only 4 weeks of training (re)uptake following trauma-related training cessation [12]. A meta-analysis demonstrated a 16.3% VO_2_max improvement in previously sedentary older adults (>60 years) of both sexes after taking up an aerobic exercise training program; the effects increased with longer training duration (i.e., >16–20 weeks) [60]. VO_2_max increases of 9–13% were observed after 8 weeks of high-intensity interval training (HIIT) in men and women of a broad age range (20 to 70+), without significant differences between age groups [61]. These findings should encourage masters athletes who become sedentary for any reason to take up training again as soon as possible. In summary, a VO_2_max loss corresponding to aging-related decline of at least one aging decade can be restored by appropriate training (re)uptake within a couple of weeks.

### 4.4. Aging-Related VO_2_max Loss despite Maintenance of High Training Volume and Intensity: The Slow Component

The age-related decline in endurance performance and its physiological determinants are largely mediated by a reduction in the volume and intensity of training commonly observed in masters endurance athletes [47]. However, as demonstrated, a VO_2_max loss of between −5% and −6.5% per decade occurred even in those who continued to engage in regular vigorous endurance exercise training and competitions [17,18,19,20]. The question arises as to which aging effects that are largely independent of training habits are responsible for this decline. In contrast to the decrease in lung function [62], both cross-sectional and longitudinal studies indicate that the decreases in maximal heart rate and stroke volume are the major contributors to the observed decline in VO_2_max in masters endurance athletes [40,47,48,63,64]. Alteration of the intrinsic heart rate may primarily account for the reduction in maximal heart rate, but diminished chronotropic responsiveness to β-adrenergic stimulation probably adds to this reduction [8,65]. In addition, lower ejection fractions at maximal exercise volume have been reported in both older sedentary and older endurance-trained athletes compared with their young peers, which is apparently related to reduced β-adrenergic responsiveness [66]. Whereas left ventricular filling pressure and arteriovenous oxygen difference remain almost unchanged at peak exercise, chronotropic and inotropic reserve (and probably Frank–Starling reserve) seem to decline even with healthy aging [67]. Whereas longitudinal and radial contributions to stroke volume do not seem to differ between groups of varying ages and training status, differences in longitudinal pumping were observed between young sedentary and old highly trained athletes compared to old sedentary individuals; the incidence of left ventricular atrioventricular plane displacement was higher in the former groups [68]. Moreover, slight macrovascular and microvascular dysfunctions accompanied by stiffening of central elastic arteries and impaired peripheral endothelial function are aging-related effects that negatively affect blood flow and oxygen delivery to exercising skeletal muscles [69]. Although endothelial function may be preserved in the leg muscles of life-long physically active people due to elevated systemic nitric oxide bioavailability, this did not restore the aging-related decline in hyperemic response [70,71]. No or only negligible impairments were reported in masters athletes with regard to muscle fiber morphology and capillary supply of the muscle tissue [72].

### 4.5. Aging-Related VO_2_max Decrease or Increase with Training Cessation or (Re)Uptake: The Fast Component

Age-related VO_2_max decline is determined by multiple factors, including reductions in blood volume, maximal heart rate and cardiac output; increased stiffening of the arterial walls; exercise-induced arterial hypoxemia; and diminished peripheral oxygen extraction. However, both loss and improvement of VO_2_max due to recent changes in exercise training seem to be primary consequences of changes (decrease or increase, respectively) in maximal cardiac output [11,22,31]. In a study on young male athletes, within the first 3 weeks, training cessation elicited a rapid response that involved a decrease in oxygen delivery due to reduced maximal cardiac output, followed by an increasing contribution of elevated arteriovenous oxygen difference [22] (Table 2). Maximal cardiac output seems to largely be a consequence of the reduction in plasma volume related limiting ventricular filling and maximal stroke volume [28,73]. The reduced stroke volume is incompletely compensated by a slight increase in maximal heart rate [28,73]. Whereas the arteriovenous oxygen difference may more rapidly recover in young and healthy males during training (re)uptake, increasing maximal cardiac output seems to contribute more importantly in older males [31]. Short-term training in older men and women results in rapid elevation of plasma volume and associated increases in cardiac output and VO_2_max [74]. This expansion of plasma volume following short-term exercise training (e.g., 2–4 weeks) has been documented in both cross-sectional and longitudinal studies [75,76]. Thermal and non-thermal components have been suggested to contribute to the elevation of plasma levels of electrolytes and proteins [75,76].

As shown in the study by Coyle et al., muscle capillarization did not markedly change over the 12-week training cessation (but remained higher than in the sedentary subjects); thus, the authors suggested that the decreasing arteriovenous oxygen difference could be attributed to a loss of mitochondria and/or mitochondrial function [22]. However, as muscle mitochondrial respiration is submaximal at VO_2_max, this likely is not the cause of VO_2_max impairment after short-term training cessation [77,78]. Calculation of the arteriovenous oxygen difference using the Fick equation suggests its reduction may result from elevated mixed venous oxygen content caused by distribution deficits of blood flow, i.e., reduced blood flow within working skeletal muscles. Skeletal muscle blood flow is diminished during dynamic exercise in older and rather sedentary individuals, resulting from a reduced vascular conductance (impaired functional sympatholysis), but improves with exercise training [79,80]. A demonstrated improvement in sympatholysis resulting from short-term exercise training was modulated by training intensity and was mediated by a nitric-oxide-dependent mechanism [81]. Moreover, elevated sympathetic vasoconstrictor responsiveness during exercise was demonstrated in older men but blunted leg vasodilator responsiveness in older women [80]. As indicated by prospective training studies, muscle sympathetic nerve activity is reduced after exercise training [82].

Thus, changes in vascular conductance due to short-term training cessation and training (re)uptake may be involved in the difference in arteriovenous oxygen response to VO_2_max changes, whereas long-term training reduction/cessation may lead to more profound remodeling of capillarization and the loss of mitochondrial content and/or efficiency.

Angiogenesis occurs as a consequence of appropriate exercise stimuli in order to increase oxygen diffusion and to improve the removal of metabolites within the contracting muscles [83]. Aging-related changes in the ultrastructure of the endothelium and the associated impairment of microcirculation are associated with a reduced CRF, but the adaptability of microcirculation to exercise stimuli seems to be maintained in old age [84,85]. Aging-related mitochondrial dysfunctions [86] and associated oxidative stress [87] are well-established hallmarks of aging. Conversely, exercise has been shown to preserve mitochondrial health in skeletal muscle [88,89,90] and slow down aging-related deterioration of antioxidative defense systems [91].

Despite the scarcity of longitudinal studies on the CRF of masters athlete, the available data indicate that reduced oxygen delivery to working muscles, mainly due to diminished cardiac output (and possibly also due to maldistribution of cardiac output), plays a major role until late middle age, and a decline in skeletal muscle oxidative capacity, at least partly due to mitochondrial dysfunction, may become increasingly important in older age (i.e., above 70 years) [92].

Fast (training cessation or (re)uptake) and slow (aging per se) components potentially modulating the VO_2_max decline in masters endurance athletes are schematically depicted in Figure 3.

## 5. How to Support the Maintenance of Sufficient Training Stimuli in Aging Masters Athletes

Training characteristics (e.g., training volume, intensity and frequency) are undisputedly important modifiers of VO_2_max in both young and older athletes. The ability to maintain a high level of exercise-training stimulus with aging seems to represent the most important measure with respect to limiting the rate of decline in VO_2_max and related endurance performance [93]. Training cessation and the associated CRF decrease may often be sequelae of disease, injuries [94] and/or motivational changes across the athletic lifespan [95]. Here, we do not focus on these aspects but solely consider physiological consequences of training cessation or training (re)uptake with respect to the aging-related CRF decline.

The decline in VO_2_max is considerably mediated by a reduction in exercise “stimuli”. Exercise stimuli are characterized by several components, including exercise-training intensity, session duration and training frequency. Several suggestions regarding effective training methods specifically for masters runners have been published [96]. Similar to younger endurance athletes [97], a combination of continuous moderate-intensity training (MIT) and high-intensity interval training (HIIT) generally appears to be efficient for masters runners. Running energy cost is largely determined by peripheral adaptations (capillarization and mitochondrial density) mainly induced by training volume/mileage at low intensity [98,99], which may explain why masters runners with several years of practice are known to be more economical than their younger counterparts [100,101]. The rate of capillarization induced by the same training program appears similar between young (22 years) and old (69 years) individuals [102]. Therefore, such low-intensity training may be less important for masters. In contrast, HIIT is associated with a well-established elevation of activity of several mitochondrial enzymes, mitochondrial biogenesis and increased levels of respiratory control [103], as first demonstrated by the pioneering work of Holloszy [104]. Various types of high-intensity training (e.g., HIIT and sprint interval training (SIT)) have since been linked to more specific mitochondrial adaptations, including changes in the phosphorylation state of mitochondria-related signaling proteins (CaMKII or AMPK), alterations in gene expression of regulators of mitochondrial function and biogenesis (e.g., PPAR; PGC-1α) or mitochondrial protein synthesis rates [105]. Moreover, the activity of numerous mitochondrial enzymes (including the mentioned citrate synthase and succinate dehydrogenase) and oxidative phosphorylation capacity are likely enhanced.

Overall, it appears that masters profit most from performing a majority of their training at high intensity (e.g., at severe or supramaximal intensities) when compared to their younger counterparts. However, this assumption can be challenged for several reasons. First, high training volume at low intensity is necessary to maintain the benefits induced by HIIT [106], with an excessive amount of HIIT sessions possibly being detrimental for mitochondrial biogenesis [107]. Secondly, the “classical” type of HIIT (e.g., 2–3 min intervals at 100% vVO_2_max) may not be optimal for older individuals due to slower VO_2_ kinetics in association with aging [108]. Whereas the time constant of the primary phase for exercise at moderate intensity averaged 50–70 s in healthy old (age 60–80 years) individuals, it amounted to only ∼30 s in young (20–30 years) healthy populations [109]. Slower muscle oxygen delivery with aging has direct consequences with respect to the prescription of HIIT, as the speed of VO_2_ kinetics is important to determine the interval duration (i.e., fast kinetics allow for short intervals, whereas slower kinetics require longer work intervals) [110]. Masters athletes are recommended to perform HIIT with longer interval durations (>5 min in the severe-intensity domain, i.e., ~90% vVO_2_max) than for younger athletes (∼3 min at vVO_2_max). Thirdly, intermittent submaximal (e.g., 15 s runs with hard bouts at 90 or 100% of vVO_2_max and an average velocity equal to 85%) sessions of varying amplitudes were also effective in masters runners, as the duration spent at VO_2_max was up to 14 min [111]. This type of session leads to considerable stimulation of the cardiovascular system combined with velocities sustainable by older individuals but high enough to stimulate neuromuscular functions. Fourthly, the need for appropriate training stimuli to counteract age-induced loss of muscle mass and strength (i.e., sarcopenia) is well documented [112]. There is no doubt that exercise should be considered a cornerstone in the treatment of such skeletal muscle wasting. Beyond the clear benefits of resistance exercise training, which significantly improves muscle mass and strength in older persons, aerobic low-intensity exercise training attenuates sarcopenia (and cachexia) [113]. Finally, the risk of increased injuries due to high-intensity running during HIIT is not negligible [114], particularly in older runners with higher rates of knee cartilage damage [115].

Altogether, reductions in volume and/or intensity of exercise training and the resulting decline in VO_2_max and performance in aging endurance athletes can be explained by various factors, such as decreasing motivation to train and/or compete, musculoskeletal injuries, pain and/or aging-related diseases. Appropriate training modifications and specific psychological and medical support are important tools to keep training breaks reasonably short and avoid substantial decreases in VO_2_max and performance. For instance, considerable reductions in pre-competition training volume for 2–3 weeks (tapering) are known to not compromise VO_2_max and even improve performance. With regard to regular training intensities, a combination of high-volume training at low intensity (≥75% of overall training volume) with low-volume training at threshold and high-intensity interval training (≤20%) appears to be an appropriate method to optimize endurance training adaptations in most middle- and long-distance runners [116].

In summary, the controversy with respect to volume versus intensity components [117,118] of exercise training is also of interest for older athletes. Despite their age-related decline in VO_2_max relative to economy, we believe that the main principles of “polarized training” remain valid in masters athletes. However, some adaptations are required (e.g., submaximal short intermittent and/or long-interval HIIT or the addition of resistance training). Furthermore, innovative training methods based on the use of systemic [119] or localized (i.e., blood flow restriction; BFR) hypoxia are relevant complements to “traditional” exercise training in older athletes. Such approaches represent potential therapeutic solutions to attain exercise benefits; for example, continuous hypoxic training induces similar cardiovascular adaptations with lower walking velocity and mechanical load compared to similar exercises in normoxia [120,121]. BFR counteracts sarcopenia and maintains muscle mass/strength with lower loads [122]. These hypoxic methods are therefore particularly well-suited for load-compromised individuals, for example, masters endurance athletes recovering from injury [123].

Finally, delayed alterations in body composition and healthy dietary choices may crucially contribute to the impressive performance of masters athletes. A reduced prevalence of sarcopenia and superior appendicular skeletal muscle mass in old age was observed in former Tokyo 1964 Olympic athletes, especially among those who continued their exercise habits with high exercise intensity [124]. In addition, the maintenance of sufficient energy availability (>30 kcal/kg LBM/day) and the ingestion of more protein-based foods (≥30 g per meal) is required to prevent the development of sarcopenia in aging athletes. This is particularly important during periods of increased training volume or, vice versa, during periods of immobilization. However, protein ingestion not only supports muscle protein synthesis but also provides amino acids, which are essential for an athlete’s recovery and adaptation. The main finding of a recent study of masters marathon runners was that protein intake per LBM, both during the tapering period and the race, was highly related to race performance, acute race-induced changes in body composition (especially of LBM) and selected metabolic and muscle-damage-related blood markers [125]. Thus, it can be assumed that the effect of protein intake during the preparation period and the race may be of even greater importance in aging endurance athletes than the amount of carbohydrate intake. Although there is little evidence that masters athletes metabolize dietary protein differently than younger athletes, endurance athletes should focus on a balanced distribution of moderate protein-containing, nutrient-dense meals throughout the day and consume as much as 0.5 g/kg immediately after exercise to replenish amino acid oxidative losses. [126]

## 6. Limitations

At least three major limitations have to be considered with regard to this review: (1) the number of longitudinal studies with precise details on training characteristics was low, (2) female masters athletes were scarcely studied and (3) mechanisms responsible for VO_2_max changes in masters athletes of both sexes remain to be elucidated. Furthermore, findings on the (re)uptake of training are predominantly derived from non-masters athletes.

## 7. Conclusions

Training reduction or cessation leads to an accelerated VO_2_max decline, as compared to the gradual aging-related VO_2_max decrease. This can rapidly nullify many of the benefits of preceding long-term training efforts. Conversely, resuming exercise training has the potential to quickly restore the entire or at least parts of the lost VO_2_max, exercise performance and health status. Interesting case studies are available to support the assumption that regular training or a return to exercise are effective for maintaining a high level of cardiovascular fitness. For example, an elite marathon runner (ex-Olympian athlete who retired from running at 32 years old with a best performance of 2:13:59 and retired from running for a 16-year period) who ran a marathon in 2:30:15 at the age of 59 [127]. Another such examples is a 70-year-old male runner with a marathon performance of 2:54:23 [128].

Despite the pronounced effects of training cessation and reuptake in the general population and masters athletes, as well as the substantial public health consequences, longitudinal studies are scarce, and the mechanistic underpinnings are largely unexplored. Future well-designed studies evaluating the time course of CRF alterations related to various training modifications and associated changes in risks and benefits with respect to the health of aging endurance athletes are urgently needed and will provide information with respect to how to expand health spans of aging athletes and non-athlete individuals.

## Figures and Tables

**Figure 2 ijerph-19-11050-f002:**
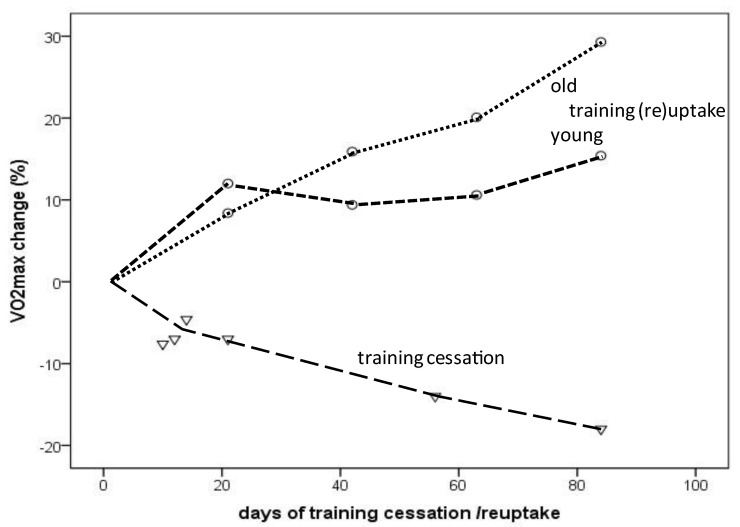
VO_2_max change depending on the amount (days) of training cessation (triangles) or training (re)uptake (circles: dotted line indicates older individuals; dashed line indicates young individuals) (from selected longitudinal data [22,24,28,31] presented in Table 2).

**Figure 3 ijerph-19-11050-f003:**
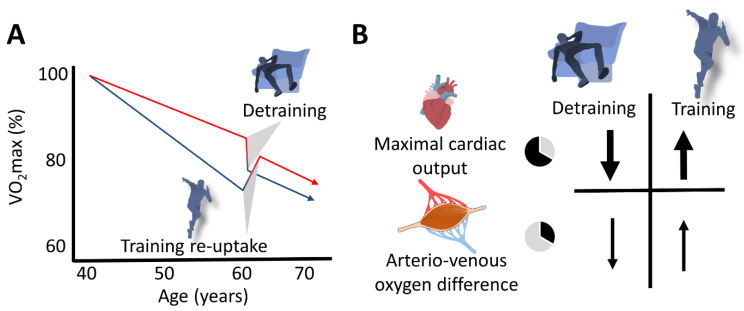
Schematic representation of the slow (aging) and the fast (changes in training habits) components of aging in masters endurance athletes. (**A**) The red line represents the aging-related decline in VO_2_max of continuously active endurance athletes, and the lower (blue) line represents that of masters athletes who terminated their competitive activities at the age of 40 years. Detraining rapidly enhances the rate of VO_2_max decline in active athletes, and recovery of VO_2_max can rapidly be achieved after (re)uptake of training. (**B**) Decline in maximal cardiac output represents the main explanatory mechanism for both the loss and restoration of VO_2_max.

**Table 2 ijerph-19-11050-t002:** Effects of short-term (up to 12 weeks) training cessation and training (re)uptake on VO_2_max and related changes of physiological parameters (cardiorespiratory parameters are maximal values).

Reference	N m (Males) f (Females)	Duration (Days) of Training Cessation (ce) Training (Re)Uptake (re)	Age (Years)	VO_2_max (mL/min/kg) Pre vs. Post (% Change)	Changes in Related Physiological Parameters (% Change)
Coyle et al., 1984 [22]	6 m 1 f	12 (ce)	29.1 ± 3.2	62.1 ± 3.3 vs. 57.7 ± 2.6 (−7%)	heart rate (+4%) stroke volume (−10%) cardiac output (−7%) arteriovenous O_2_diff (+0.4%) oxygen pulse (−11%) citrate synthase (−17.1%) succinate dehydrogenase (−18.5%)
Coyle et al., 1984 [22]	6 m 1 f	21 (ce)	29.1 ± 3.2	62.1 ± 3.3 vs. 59.7 ± 3.1 (−7%)	heart rate (+4%) stroke volume (−11%) cardiac output (−8%) arteriovenous O_2_diff (+2%) oxygen pulse (−11%) citrate synthase (−23.7%) succinate dehydrogenase (−23.9%)
Coyle et al., 1984 [22]	6 m 1 f	56 (ce)	29.1 ± 3.2	62.1 ± 3.3 vs. 53.2 ± 2.1 (−14%)	heart rate (+6%) stroke volume (−14%) cardiac output (−9%) arteriovenous O_2_diff (−4%) oxygen pulse (−19%) citrate synthase (−40.6%) succinate dehydrogenase (−38.4%)
Coyle et al., 1984 [22]	6 m 1 f	84 (ce)	29.1 ± 3.2	62.1 ± 3.3 vs. 50.8 ± 1.9 (−18%)	heart rate (+5%) stroke volume (−13%) cardiac output (−10%) arteriovenous O_2_diff (−7%) oxygen pulse (−20%) citrate synthase (−39.6%) succinate dehydrogenase (−32.5%)
Cullinane et al., 1986 [23]	15 m	10 (ce)	28.2 ± 5.6	61.3 ± 6.2 vs. 61.2 ± 5.6 (−1.6%)	heart rate (+5%) ventilation (+1.5%) stroke volume (−2.6%)
Doherty et al., 2003 [24]	7 f	10 (ce)	21.0 ± 2.6	49.8 ± 1.3 vs. 46.0 ± 1.3 (−7.6%)	heart rate (+1.5%) stroke volume (−1%) cardiac output (−0.5%) arteriovenous O_2_diff (−7%)
Drinkwater and Horwath, 1972 [25]	7 f	90 (ce)	14-17	47.8 ± 1.8 vs. 40.4 ± 1.0 (−15.4%)	heart rate (+1.5%) ventilation (−10.3%)
Giada et al., 1998 [26]	12 m	60 (ce)	55 ± 5	43 ± 7 vs. 36 ± 7 (−16.3%)	
Giada et al., 1998	12 m	60 (ce)	24 ± 6	59 ± 10 vs. 49 ± 9 (−16.9%)	
Heath et al., 1983 [27]	6 m 2 f	10 (ce)	28 ± 3	58.6 ± 2.2 vs. 57.6 ± 2.1 (−1.7%)	
Houmard et al., 1992 [28]	9 m 3 f	14 (ce)	20.1 ± 1.4	61.6 ± 2.2 vs. 58.7 ± 1.8 (−4.6%)	heart rate (+4.7%) plasma volume (−5.1%) citrate synthase (−25.3%)
Katzel et al., 1997 [29]	10 m	90 (ce)	59 ± 8	50 ± 5 (−11 to −20%)	
Martin et al., 1986 [30]	5 m 1 f	42 (ce)	26 ± 1	62.7 ± 4.0 (−6.5%), 21 days; (−20.3%), 56 days	stroke volume 21 days (−10%) 56 days (−17%)
Murias et al., 2010 [31]	8 m	21 (re)	68.0 ± 7.0	28.3 ± 7.1 vs. 30.7 ± 6.0 (+8.4%)	heart rate (−3.5%) stroke volume (+6.8%) cardiac output (+7%) arteriovenous O_2_diff (+3.7%)
Murias et al., 2010 [31]	8 m	21 (re)	23.0 ± 5.0	48.0 ± 6.1 vs. 53.8 ± 7.6 (+12%)	heart rate (−2.1%) stroke volume (+5.5%) cardiac output (+3.1%) arteriovenous O_2_diff (+7.5%)
Murias et al., 2010 [31]	8 m	42 (re)	68.0 ± 7.0	28.3 ± 7.1 vs. 32.8 ± 7.6 (+15.9%)	heart rate (−2.1%) stroke volume (+9.1%) cardiac output (+11.3%) arteriovenous O_2_diff (+5.2%)
Murias et al., 2010 [31]	8 m	42 (re)	23.0 ± 5.0	48.0 ± 6.1 vs. 52.5 ± 6.4% (+9.4%)	heart rate (−2.1%) stroke volume (+7.9%) cardiac output (+5.4%) arteriovenous O_2_diff (+4.8%)
Murias et al., 2010 [31]	8 m	63 (re)	68.0 ± 7.0	28.3 ± 7.1 vs. 34.0 ± 5.8 (+20.1%)	heart rate (−1.4%) stroke volume (+15.2%) cardiac output (+17.9%) arteriovenous O_2_diff (+3.7%)
Murias et al., 2010 [31]	8 m	63 (re)	23.0 ± 5.0	48.0 ± 6.1 vs. 53.1 ± 6.5% (+10.6%)	heart rate (−2.1%) stroke volume (+12.6%) cardiac output (+10.4%) arteriovenous O_2_diff (+0.7%)
Murias et al., 2010 [31]	8 m	84 (re)	68.0 ± 7.0	28.3 ± 7.1 vs. 36.6 ± 6.5 (+29.3%)	heart rate (+0.7%) stroke volume (+14.8%) cardiac output (+20.8%) arteriovenous O_2_diff (+8.8%)
Murias et al., 2010 [31]	8 m	84 (re)	23.0 ± 5.0	48.0 ± 6.1 vs. 55.4 ± 5.5% (+15.4%)	heart rate (−1.1%) stroke volume (+10.9%) cardiac output (+9.7%) arteriovenous O_2_diff (+6.8%)
Nichols et al., 2000 [12]	1 f	14 (re)	49.4	42.0 vs. 48.1 (+14.5%)	heart rate (+1.6%)
Nichols et al., 2000 [12]	1 f	28 (re)	49.4	42.0 vs. 55.1 (+31.2%)	heart rate (−2.7%)
Pavlik et al., 1986 [32]	42 m	60 (ce)	22.9 ± 0.7	72.2 vs. 67.0 (30 days) vs. 62.5 (45 days) (−7% and −13%); no further decrease after 45 days	
Prior et al., 2015 [33]	7 m 5 f	14 (ce)	65 ± 3	31.2 ± 2.3 vs. 29.3 ± 1.9 (−6%)	citrate synthase (−28.6%)
Ready and Quinney, 1982 [34]	12 m	63 (ce)	25.0 ± 3.6	64.2 ± 9.5 vs. 59.3 ± 6.4, (3 weeks) 57.5 ± 6.4 (6 weeks) 57.3 ± 8.8 (9 weeks) (−10.7%)	
Schulman et al., 1996 [35]	8 m	84 (ce)	59.6 ± 3	49.9 ± 1.9 vs. 42.0 ± 2.2 (−15.8%)	heart rate (+4.1%) cardiac index (−10.5%) stroke volume index (−14.2%)
Sinacore et al., 1993 [36]	5 m 1 f	84 (ce)	29 ± 10	61.3 ± 7 vs. 50.8 ± 7 (−17.1%)	
Smorawinski et al., 2001 [37]	10 m	3 (ce, bedrest)	20.3 ± 1.9	54.8 ± 2.1 (−16.5%)	blood lactate (−8% to −20%)

## Data Availability

Not applicable.

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
