# Peer review of "The Impact of Training on the Loss of Cardiorespiratory Fitness in Aging Masters Endurance Athletes"

_ijerph, 2022, doi:10.3390/ijerph191711050_

Round 1
Reviewer 1 Report
Title: The impact of training on the loss of cardiorespiratory fitness in the aging Masters endurance athletes
Johannes Burtscher , Barbara Strasser , Martin Burtscher*, Gregoire P. Millet
Type of work: Review
Suggestions to the authors:
1. The authors have only used 185 words in the abstract. They could tighten the abstract. They can then add the changes of related to physiological parameters. (HR, Stroke volume, citrate synthase, SDH (mitochondrial enzymes)). I think that would increase your readership.
2. Figure 2 needs definition of data symbols. It should stand on its own and so some additionala information is need in the legend. Also solid lines should not be used to connect the dots as the data are not continuous.
REVIEW
Purpose:
The importance of maintaining appropriate training loads and in particular the consequences of s of training (re)uptake in Masters endurance athletes have not been thoroughly investigated. The authors review select publications with the aim to evaluate and summarize longitudinal observations on V̇O2max changes related to training variations in the long and short run of Masters endurance athletes. Authors included studies of both males and females. They analyzed the data by gender.
This is a descriptive study. Data gathered from publications with select criteria were analyzed by multiple linear regression. A stepwise variable selection was applied to evaluate the predictive importance of independent variables on V̇O2max changes. Training volume, age, sex, the observation period (in years), and baseline V̇O2max (at the beginning of the longitudinal observation) were considered as independent variables.
Results: Results are supported by Tabled data.
The most notable outcomes are quoted from the text as well as summarized by the reviewer.
“The V̇O2max 77 (mL/min/kg) decline per decade ranged from -5% to -46% per decade and was closely related to the changes in training volume (running, km/week). Studies reporting training intensities (running, min/km) indicate that training volume reductions parallel intensity reductions, except one of the female groups of the study by Eskurza et al., who even increased training volume but reduced training intensity during follow-up. Male Masters who almost maintained training volume (not more than 10% reduction or remained highly trained) showed a V̇O2max decline between -5% and -6.5% per decade.”
“Regression analysis including male athletes [17-21], revealed a close association 94 between reported V̇O2max reductions and related changes of training volumes with aging. Fifty-four percent of the variance of the observed V̇O2max decline was explained by training-volume changes (figure 1), and this percentage increased to 70% when the age of the athletes was considered. No other variables improved the explanation of the V̇O2max decline. Within the groups of females, 39% of the variance in V̇O2max change was explained by changes in the training volume.”
“With training (re)uptake in older individuals, the continuously increasing cardiac output with training duration contributed more importantly to the V̇O2max improvement than the arteriovenous-oxygen differences (table 2). This is in contrast to young individuals who showed a larger improvement of the arteriovenous-oxygen difference especially during the first weeks of training.”
In the present study, the association between the V̇O2max decline and changes in training volume (and intensity) is less pronounced in female compared to male endurance athletes
Though very few data on training cessation or training (re)uptake responses (physiological characteristics) in older Masters are available, the authors found enough publications to report worthwhile analyses.
The data showing reduced cellular aerobic capacity based on reduced mitochondrial content and/or efficiency is very worthwhile.
Tables are all well organized and necessary.
Figures are interesting and necessary.
Figure 2 needs definition of data symbols. It should stand on its own and so some additionala information is need in the legend. Also solid lines should not be used to connect the dots as the data are not continuous.
The Discussion is thorough. Authors successfully organized it into readable sections with topic sentences.
The authors include BONUS information in the Discussion. The paragraphs are labeled, “How to support the maintenance of sufficient training stimuli in the aging Masters athletes? This sections is especially worthwhile reading for training older adults.
The Limitations Section rings true true and will hopefully guide future studies.
The number of references culled is far more than adequate. References are publications from the most reputable laboratories which study cardiorespiratory fitness and aging. The references historically include applicable studies from 1967 to present day.
Suggestions to the authors:
1. The authors have only used 185 words in the abstract. They could tighten the abstract. They can then add the changes of related to physiological parameters. (HR, Stroke volume, citrate synthase, SDH (mitochondrial enzymes)). I think that would increase your readership.
2. Figure 2 needs definition of data symbols. It should stand on its own and so some additionala information is need in the legend. Also solid lines should not be used to connect the dots as the data are not continuous.

Author Response
Dear Reviewer: First of all, we would like to thank you very much for the favorable, comprehensive and constructive comments that really helped to improve the paper.
We tried to respond adequately to all your points (below); changes in the revised manuscript are shown in the correction mode.
Suggestions to the authors:
- The authors have only used 185 words in the abstract. They could tighten the abstract. They can then add the changes of related to physiological parameters. (HR, Stroke volume, citrate synthase, SDH (mitochondrial enzymes)). I think that would increase your readership.
Re: Thank you for this suggestion. We now added this information and state as follows: “Beside a decline in stroke volume and cardiac output, training cessation was accompanied by considerable reductions in citrate synthase and succinate dehydrogenase activity (reduction of mitochondrial content and oxidative capacity).”
- Figure 2 needs definition of data symbols. It should stand on its own and so some additional information is need in the legend. Also, solid lines should not be used to connect the dots as the data are not continuous.
Re: Thank you very much! We added the missing information and changed lines in the figure.
Reviewer 2 Report
This review paper investigated the effect of training on the loss of cardiorespiratory fitness in the aging masters endurance athletes. Overall, the review is well written, while some items need to be addressed as follows:
Introduction
Lines 44-45: the author mentioned the importance of maintaining the appropriate training loads and the training uptake in Masters endurance athletes has been much less investigated. The author only mentioned it is important, but needs to explain why it is important to make the rationale of the study more specific.
Materials and Methods
Lines 68-70: the author used volume, age, sex, the observation period, and baseline VO2max as the independent variables for the multiple linear regression model. Are these independent variables selected based on the previous research reference?
Results:
Figures 1 and 2, the author needs to explain both figures in more detail.
Author Response
Dear Reviewer: First of all, thank you very much for the favorable and really helpful comments.
We tried to respond adequately to all your points (below); changes in the revised manuscript are shown in the correction mode.
This review paper investigated the effect of training on the loss of cardiorespiratory fitness in the aging masters endurance athletes. Overall, the review is well written, while some items need to be addressed as follows:
Introduction
Lines 44-45: the author mentioned the importance of maintaining the appropriate training loads and the training uptake in Masters endurance athletes has been much less investigated. The author only mentioned it is important, but needs to explain why it is important to make the rationale of the study more specific.
Re: Thank you for this point. We now explain as follows: “However, regular and appropriate training stimuli are of utmost importance to improve or maintain muscular performance and CRF [11,12].”
Materials and Methods
Lines 68-70: the author used volume, age, sex, the observation period, and baseline VO2max as the independent variables for the multiple linear regression model. Are these independent variables selected based on the previous research reference?
Re: Yes.
Results:
Figures 1 and 2, the author needs to explain both figures in more detail.
Re: Thank you. This point has also been raised by reviewer 1. We improved as suggested and added to the legend of figure 1: “Changes in the training volume explain 54% and 39% of the variance in VO2max changes in male and female Masters athletes, respectively.” We changed figure 2 (as suggested by reviewer 1) and changed the legend to: “VO2max change depending on the amount (days) of training cessation (triangles) or training (re)uptake (circles: dotted line indicates older individuals, dashed line indicates young individuals) (from selected longitudinal data [22,24,28,31] presented in table 2)”.
Reviewer 3 Report
Dear Authors,
Thanks a lot for giving the opportunity to review this excellent manuscript. The authors are well known experts in the field and have published many relevant studies on the topic.
The manuscript is a review that deals with the impact of training on the loss of cardiorespiratory fitness in the aging Masters endurance athletes. It is the first review on this important topic, that is highly relevant to our ageing population. The results suggest that reductions in VO2max due to changes in training volume led to a VO2max drop that can quickly and largely be recovered upon (re)uptake of training. These results shed new light on the discussion on exercise benefits at higher age and underline the benefits for older people to start endurance training.
The manuscript is technically sound, the tables and figures are well done. The 128 references are appropriate for the article length and well selected. The writing is excellent – I am not a native English speaker, but I did not find a single mistake.
For these reasons, I suggest to accept the manuscript in its common form.
Author Response
Dear Reviewer: Thanks a lot for the favorable comments.
We tried to respond adequately to all points raised by the reviewers (below); changes in the revised manuscript are shown in the correction mode.
The manuscript is a review that deals with the impact of training on the loss of cardiorespiratory fitness in the aging Masters endurance athletes. It is the first review on this important topic, that is highly relevant to our ageing population. The results suggest that reductions in VO2max due to changes in training volume led to a VO2max drop that can quickly and largely be recovered upon (re)uptake of training. These results shed new light on the discussion on exercise benefits at higher age and underline the benefits for older people to start endurance training.
The manuscript is technically sound, the tables and figures are well done. The 128 references are appropriate for the article length and well selected. The writing is excellent – I am not a native English speaker, but I did not find a single mistake.
For these reasons, I suggest to accept the manuscript in its common form.
Re: Thank you very much again!